# Strained Lattice Gold-Copper Alloy Nanoparticles for Efficient Carbon Dioxide Electroreduction

**DOI:** 10.3390/ma15145064

**Published:** 2022-07-20

**Authors:** Fangfang Chang, Chenguang Wang, Xueli Wu, Yongpeng Liu, Juncai Wei, Zhengyu Bai, Lin Yang

**Affiliations:** Collaborative Innovation Center of Henan Province for Green Manufacturing of Fine Chemicals, Key Laboratory of Green Chemical Media and Reactions, Ministry of Education, School of Chemistry and Chemical Engineering, Henan Normal University, Xinxiang 453007, China; changfangfang@htu.edu.cn (F.C.); wcg0630@163.com (C.W.); shirely19991008@163.com (X.W.); liuyongpengaoe@163.com (Y.L.); wjc874380607@163.com (J.W.)

**Keywords:** lattice strain, crystal plane, Faradaic efficiency, CO_2_ reduction reaction

## Abstract

Electrocatalytic conversion of carbon dioxide (CO_2_) into specific renewable fuels is an attractive way to mitigate the greenhouse effect and solve the energy crisis. Au_n_Cu_100-n_/C alloy nanoparticles (Au_n_Cu_100−n_/C NPs) with tunable compositions, a highly active crystal plane and a strained lattice were synthesized by the thermal solvent co-reduction method. Transmission electron microscopy (TEM) and X-ray diffraction (XRD) results show that Au_n_Cu_100−n_/C catalysts display a subtle lattice strain and dominant (111) crystal plane, which can be adjusted by the alloy composition. Electrochemical results show that Au_n_Cu_100−n_/C alloy catalysts for CO_2_ reduction display high catalytic activity; in particular, the Faradaic efficiency of Au_75_Cu_25_/C is up to 92.6% for CO at −0.7 V (vs. the reversible hydrogen electrode), which is related to lattice shrinkage and the active facet. This research provides a new strategy with which to design strong and active nanoalloy catalysts with lattice mismatch and main active surfaces for CO_2_ reduction reaction.

## 1. Introduction

Converting CO_2_ into useful chemicals can reduce the concentration of CO_2_ in the atmosphere and realize the recycling of CO_2_, which has attracted extensive attention of researchers. Electrochemical reduction reaction of CO_2_ (CO_2_RR) can be performed using electricity produced by renewable energy sources, such as wind energy and hydropower [1,2,3]. There have been a number of reports on catalysts for the electroreduction of carbon dioxide [4]. In this regard, noble metals have proved to be promising catalysts for the electroreduction of CO_2_ to CO [5,6,7,8,9]. A large number of studies have shown that metal nanostructured catalysts with more active sites, such as Au, Ag and Pd, can greatly improve the catalytic activity of CO_2_ reduction [5,10,11,12,13,14,15]. However, the multiple electron transfer in the CO_2_RR process, the reaction pathway, the hydrogen evolution reaction (HER) and the high price and low reserves of noble metals hinder the wide application of precious metals [16,17,18]. Therefore, reducing catalyst costs and improving activity and selectivity are challenges [19]. Many studies have shown that the incorporation of cheap metals into noble metal nanocrystals can also improve their catalytic performances and efficiency. The formation of the alloy changes the lattice of the metal catalyst, resulting in lattice strain [20,21]. The lattice strain generated by the catalyst can change the electronic properties of the metal and improve the electrocatalytic activity [22]. Previous studies have shown that Au-based catalysts have high activity and selectivity for CO_2_ reduction to CO [23]. Introducing cheap metals with Au to form alloy catalysts is necessary to improving the catalytic activity for CO_2_ reduction and decreasing the price of catalyst [24,25]. The composition of the alloy catalyst can be adjusted to regulate the electronic structure and enhance the catalytic activity and stability [26]. Gold and copper form nano-alloys in different proportions, and lattice strains are controlled by changing the proportions of gold and copper in the material [27]. Previous studies have found that the lattice spacing of the alloy expands and contracts to varying degrees with the changes in the atomic proportions of the two metals [28,29]. The expansion and contraction of lattice spacing are the key factors affecting the electrochemical properties of the alloy [30,31].

In this work, the thermal solvent co-reduction method was used to synthesize Au_n_Cu_100−n_/C NPs with tunable compositions, a highly active crystal plane and a strained lattice. TEM results show that Au_n_Cu_100−n_/C catalysts display a dominant (111) crystal plane. XRD results show that the lattice constant shrinks and expands through tuning the compositions of catalysts. Electrochemical results show that Au_n_Cu_100−n_/C alloy catalysts for CO_2_ reduction display highly catalytic activity; in particular, the Faradaic efficiency of Au_75_Cu_25_/C is up to 92.6% for CO at −0.7 V vs. RHE, which is related to lattice shrinkage and bimetallic compositions.

## 2. Materials and Methods

### 2.1. Chemicals

Hydrogen tetrachloroaurate (III) hydrate (HAuCl_4_·xH_2_O, 49%~51% Au basis), copper dinitrate (Cu(NO_3_)_2_, AR), sodium hydroxide (NaOH, AR), ethylenediamine, hydrazine (80%), sodium thiosulphate (Na_2_S_2_O_3_, AR), deionized water, Nafion (5 wt%) and ethanol (99.7%) were obtained from Deen reagent. Carbon black (Vulcan XC-72) was purchased from Cabot. All gases were obtained from Airgas. All chemicals were used without further purification.

### 2.2. Preparation Cu NPs

The Cu NPs were prepared by a simple method. Cu(NO_3_)_2_ (376.0 mg) and NaOH (8.0 g) were dissolved in 20 mL deionized water to form a uniform solution. Then, 4 mL ethylenediamine and 1 mL hydrazine were added to the above solution. After all the reactants were thoroughly mixed and transferred to a flask, it was placed in a water bath at 80 °C for 1 h. Finally, the product was washed four times with deionized water and ethanol to obtain Cu NPs [32].

### 2.3. Preparation Au_n_Cu_100−n_ NPs

Cu NPs (64.0 mg) and Na_2_S_2_O_3_ (79.0 mg) were dissolved in 100 mL deionized water saturated with N_2_ and dispersed by ultrasonication. When Cu NPs were completely dispersed, 340.0 mg HAuCl_4_·xH_2_O was added, and the reaction was carried out under magnetic stirring for 30 min. Finally, the product was collected by centrifugation, washed four times with ethanol and dried under vacuum to obtain Au_50_Cu_50_ NPs. Au_25_Cu_75_ NPs and Au_75_Cu_25_ NPs catalysts were prepared under similar conditions where n(HAuCl_4_ × xH_2_O):n(Cu(NO_3_)_2_) was 1:3 or 3:1, respectively. All the catalysts were loaded onto the carbon black to obtain Au_25_Cu_75_/C, Au_50_Cu_50_/C and Au_75_Cu_25_/C.

### 2.4. Characterizations

Transmission electron microscopy (TEM) and high-resolution TEM (HR-TEM) were used to characterize the morphology and size of Au_n_Cu_100−n_/C catalysts performed on JEM-2100F TEM working at 200 kV [33]. The structures of Au_n_Cu_100−n_/C were measured on a Shimadzu X-ray diffractometer (XRD) instrument operating with Cu Kα (λ = 0.154 nm) radiation [34]. X-ray photoelectron spectroscopy (XPS) can determine the content and chemical states of the elements contained on the surface of a sample [35,36].

### 2.5. Electrochemical Measurements

To prepare a working electrode for electrochemical activity test, 2 mL of a mixture containing deionized water, isopropanol and Nafion (5% wt) (9:1:15, *V*/*V*/*V*) was ultrasonic dispersed on a 4 mg Au_n_Cu_100−n_/C powder catalyst for 60 min to form a homogeneous catalyst ink (2 mg/mL). The prepared catalyst suspension (300 μL) was coated on the surface of carbon paper with an area of 1 cm × 1 cm [37]. Electrocatalytic reduction of CO_2_ was performed on a computer-controlled electrochemical analyzer (CHI760e, CH Instruments). All the experiments were conducted in a gas-tight H-type cell with cathode and anode compartments separated by a Nafion^®^ NRE-212 proton exchange membrane. The H-type cell was filled with a 0.1 M KHCO_3_ solution (pH = 6.8, 45 mL) as the electrolyte in each chamber with 15 mL headspace. Platinum foil and Ag/AgCl (saturated KCl) were used as a counter electrode and reference electrode. The pH of the electrolyte was measured by the Thermo Scientific Orion Versa Star pH Benchtop Tester (INESA). In the process of electrochemical reduction of CO_2_, the mass flow controller (Sevenstar, Beijing) was used to purge CO_2_ at the flow rate of 20 mL/min. Before each electrochemical experiment, CO_2_ was purged into the cathodic compartment for at least 40 min until the solution pH reached 6.8 (CO_2_-saturated 0.1 M KHCO_3_). The working electrode was activated by cyclic voltammetry (CV) until a stable curve at room temperature and ambient pressure. The gas products in the cathode chamber were quantitatively analyzed by an online gas chromatograph (GC2030, Shimadzu) equipped with a thermal conductivity detector (TCD) and flame ionization detector (FID) [38]. The Faraday efficiency (FE) was calculated by dividing the amount of charge transferred to the gas product by the total amount of charge transferred in a specific time or the entire reduction reaction (for gas products). In this work, the potentials were adjusted to reversible hydrogen electrode (RHE) potentials. The electrochemical active area (ECSA) was obtained from a cyclic voltammogram in 50 mM H_2_SO_4_ [39].

## 3. Results and Discussion

### 3.1. Morphology

Au_n_Cu_100−n_ NPs was prepared by galvanic replacement reaction assisted by Na_2_S_2_O_3_ in an aqueous solution composed of HAuCl_4_ and copper NPs, where Na_2_S_2_O_3_ acted as an inhibitor of CuCl disproportionation caused by Cu/HAuCl_4_ substitution. The compositions of Au_n_Cu_100−n_ NPs were controlled by adjusting the ratios of Cu NPs and HAuCl_4_. The morphology and alloy structure of the Au_n_Cu_100−n_ NPs were obtained by TEM and HR-TEM. The compositions of the Au_n_Cu_100−n_ NPs were controlled by the metal precursor ratios and analyzed by ICP-MS, indicating that the compositions of the Au_n_Cu_100−n_ NPs can be controlled well by tuning the feeding ratio during the synthesis. It can be seen in Figure 1 that the diameter of Au_25_Cu_75_ NPs was 10–16 nm (Figure 1a), that of Au_50_Cu_50_ NPs was 8–12 nm (Figure 1b) and that of Au_75_Cu_25_ NPs was 6–10 nm, via TEM (Figure 1c). It can be concluded from the TEM images that with the increase in Au content in Au_n_Cu_100−n_ alloy NPs, the diameter of Au_n_Cu_100−n_ alloy NPs gradually decreased. HR-TEM images revealed that Au_n_Cu_100−n_ NPs are a dominant (111) facet and the continuous lattice fringes of Au_n_Cu_100−n_ NPs calculated were slightly smaller than those of the pure Au (0.235 nm) and larger than that of pure Cu (0.208 nm), indicating that Au successfully replaced Cu to form AuCu alloy catalysts, as shown in Figure 1d–f. The Au_n_Cu_100−n_ alloyed structure was also achieved by the elemental mapping technique (Figure 1g–i) [32], which showed that Au and Cu were evenly distributed in the Au_n_Cu_100−n_ NPs.

### 3.2. Structures

The differences in crystalline structures between Au_n_Cu_100−n_/C NPs were determined from X-ray diffraction (XRD). As shown in Figure 2a, the peak positions located at 38.2°, 44.4° and 64.9° diffractions further confirmed the formation of an alloy system, as they are located between those of pure Au and pure Cu. Figure 2b shows that the lattice spacing was basically linear in accordance with Vegard’s law, but there was a slight deviation, which mainly depended on the compositions of Au_n_Cu_100−n_/C. The lattice constant shrunk when the Au% was more than 50%, but when the Au% was less than 50%, the lattice spacing showed lattice expansion. The compositions, structures and valence states on the surface of catalysts were further analyzed by XPS spectra. Figure 2c–d show the Au 4f region (Au 4f_5/2_ and Au 4f_7/2_) and Cu 2p region (Cu 2p_1/2_ and Cu 2p_3/2_), corresponding to Au^0^ and Cu^0^ chemical states, respectively. However, the positive shift of the Au 4f spectrum occurred with the increase in Au%, especially for the Au_75_Cu_25_/C composition, indicating that the catalyst had a partial positive charge, and the transition of the d-band center, which is usually regarded as an effective descriptor for evaluating the catalytic activity and is beneficial for electrochemical catalysis. The adsorption and desorption capacity of the reaction product on the catalyst surface is closely related to the binding energy, which indicates that the center of the d band moves downward in the Au_n_Cu_100−n_/C catalyst compared with pure Au. The electrons from Cu to Au in the Au_n_Cu_100−n_ NPs catalyst doubled the local electron density around the Au sites, which has been shown to reduce the intermediate CO generated during the catalytic process and the adsorption of catalyst poisons to prevent catalytically active sites’ formation.

### 3.3. Electrochemical CO_2_ Reduction on the Au_n_Cu_100−n_/C Catalysts

The electrocatalytic properties of Au_n_Cu_100−n_/C for CO_2_RR were evaluated in a gas-tight H-type electrolyzer in 0.1 M KHCO_3_ electrolyte saturated with Ar or CO_2_ under room temperature and standard atmospheric pressure. For quantitative analysis of the gaseous products, the H-type electrolyzer was connected directly with the gas chromatograph (GC) at the gas outlet. The cyclic voltammetry (CV) and linear sweep voltammetry (LSV) curves of the Au_n_Cu_100−n_/C catalyst were tested in CO_2_ and Ar-saturated 0.1 M KHCO_3_ electrolyte, respectively (Figure 3, Figure 4 and Figure 5). The LSV of three different composition catalysts in CO_2_-saturated 0.1 M KHCO_3_ electrolyte are given in Figure 6a. Apparently, Au_75_Cu_25_/C exhibited higher total current density than the other catalysts, suggesting that Au_75_Cu_25_/C was the most active catalyst for CO_2_ reduction. Moreover, the cathodic current densities of the three catalysts in CO_2_-saturated electrolyte were all higher than in the Ar-saturated electrolyte, indicating that CO_2_RR occurred. Furthermore, the electrochemically active surface area (ECSA) of the Au_n_Cu_100−n_ alloy was tested to verify the catalytic activity. The results are shown in Figure 7. Au_75_Cu_25_/C had the largest electrochemically active surface area. The electrochemically active surface area of Au_75_Cu_25_/C was 25.2 m^2^ g^−^^1^, greater than those of Au_25_Cu_75_/C (6.38 m^2^ g^−^^1^) and Au_50_Cu_50_/C (17.6 m^2^ g^−^^1^). The value of the electrochemically active surface area was increased with Au content, which is related to the facet and lattice strain of catalysts. In the process of CO_2_ reduction, the gas products produced in the electrolytic cell were analyzed every 30 min by the GC sampling system at each given potential. Figure 6b shows the FE of CO for different component catalysts in the CO_2_RR process. It can be seen in Figure 6b and Figure 8 that the gas products only included CO and H_2_. The FE_CO_ of Au_75_Cu_25_/C was the highest at −0.7 V vs. RHE, achieving 92.6%. As shown in Figure 6c, the current density of Au_75_Cu_25_/C for CO was 28.5 mA cm^−^^2^ at −0.7 V vs. RHE, which is several times the values of other catalysts.

For comparisons with the pure metal catalysts, we used the same method to synthesize Au NPs and Cu NPs, and they were tested for CO_2_RR, as shown in the Figure 6, Figure 9 and Figure 10, respectively. The results show that the activity and selectivities of the two catalysts are lower than those of the alloy catalysts.

Stability is an important indicator with which to evaluate the performances of electrocatalysts. The stability of Au_75_Cu_25_/C was tested by the timing current method (i–t), and the gas products were detected with an online gas chromatograph every hour for 10 h (Figure 6d). In the reaction process for 10 h, the catalytic activity of Au_75_Cu_25_/C catalyst for CO had no obvious attenuation, and FE_CO_ remained above 90%. This indicates that Au_75_Cu_25_/C maintained good catalytic activity and selectivity for CO in the CO_2_RR process, and could keep stable. The results show that alloying caused changes in lattice parameters between catalysts which affected the performance of the catalysts for the CO_2_RR. In order to further prove that Au_75_Cu_25_/C catalyst has good stability, XPS was used to characterize the Au_75_Cu_25_/C catalyst after electrolysis. XPS results show that Au_75_Cu_25_/C retained its original composition after a long period of electrolysis (Figure 11). In the process of electroreduction for CO_2_, Au and Cu cooperated with each other to form an alloy and showed high catalytic activity and selectivity for CO. Meanwhile, the unique nanostructure of Au_n_Cu_100−n_ alloy could reduce the adsorption of CO on the surface and increase the yield of CO. The high selectivity of this gas product makes the separation process relatively simple, which is beneficial for practical applications.

## 4. Conclusions

In summary, an Au_n_Cu_100−n_ alloy dominant (111) facet was synthesized with a simple method. The composition of Au_n_Cu_100−n_ alloy can be simply controlled by adjusting the Au/Cu atomic ratio of the precursor. Na_2_S_2_O_3_ and NaOH as inhibitors that also prevent NPs aggregation play important roles in the synthesis of Au_n_Cu_100−n_. Au_75_Cu_25_/C exhibits excellent catalytic activity and stability for CO_2_ electroreduction. The excellent catalytic performance of Au_75_Cu_25_/C mainly depends on the large specific surface area and inter-lattice shrinkage. Meanwhile, the coordination of Au and Cu in the catalyst is also a reason for its good stability and the good performance of the catalyst. Au_75_Cu_25_/C had the highest Faradaic efficiency and the highest CO selectivity compared with pure gold catalysts in the CO_2_ reduction process. Our study on Au_n_Cu_100−n_ alloy catalysts demonstrated that the formation of alloys of noble metals with inexpensive metals can improve the activity and selectivity of the catalysts, which provides a new strategy for the design of catalysts for CO_2_RR.

## Figures and Tables

**Figure 1 materials-15-05064-f001:**
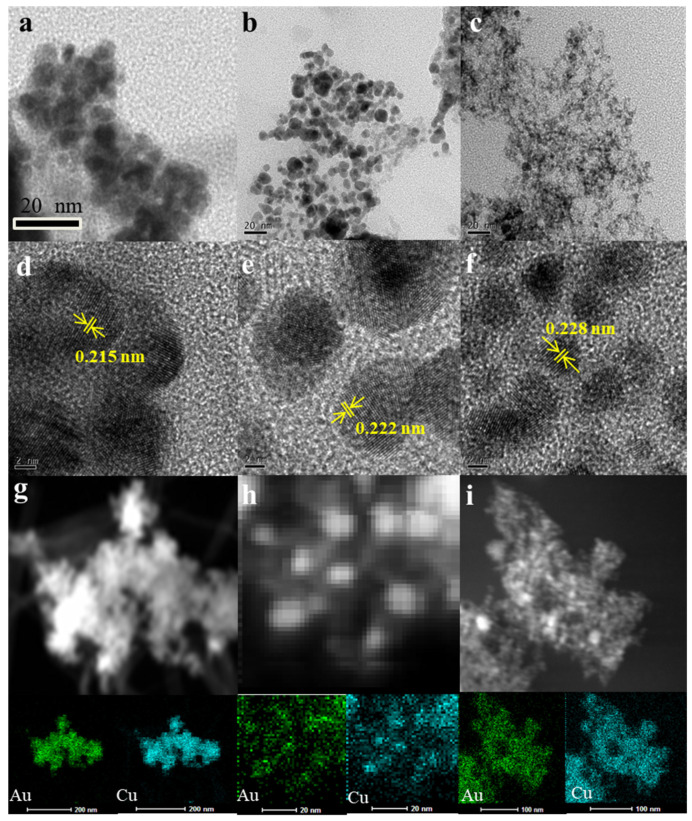
TEM and HR-TEM images of the Au_n_Cu_100−n_ NPs samples. Au_25_Cu_75_ (**a**,**d**), Au_50_Cu_50_ (**b**,**e**) and Au_75_Cu_25_ (**c**,**f**) with lattice fringes and corresponding facets indicated. EDS element mappings of Au_25_Cu_75_ (**g**), Au_50_Cu_50_ (**h**) and Au_75_Cu_25_ (**i**).

**Figure 2 materials-15-05064-f002:**
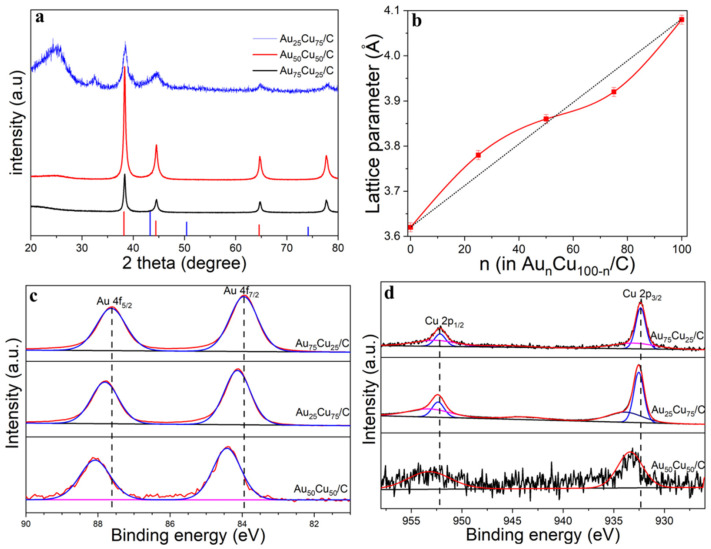
(**a**) XRD patterns of Au_25_Cu_75_/C, Au_50_Cu_50_/C and Au_75_Cu_25_/C. (**b**) Dependence of the lattice parameters for the Au_n_Cu_100−n_ NPs samples on the relative composition of Au%. (**c**) Au 4f XPS spectra of Au_25_Cu_75_/C, Au_50_Cu_50_/C and Au_75_Cu_25_/C. (**d**) Cu 2p XPS spectra of Au_25_Cu_75_/C, Au_50_Cu_50_/C and Au_75_Cu_25_/C.

**Figure 3 materials-15-05064-f003:**
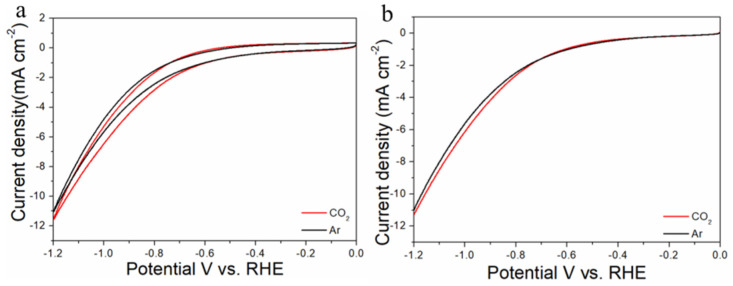
Cyclic voltammograms (**a**) and linear scan voltammograms (**b**) of Au_25_Cu_75_/C in Ar saturated and CO_2_-satruated 0.1 M KHCO_3_ solutions collected at a scan rate of 20 mV s^−^^1^.

**Figure 4 materials-15-05064-f004:**
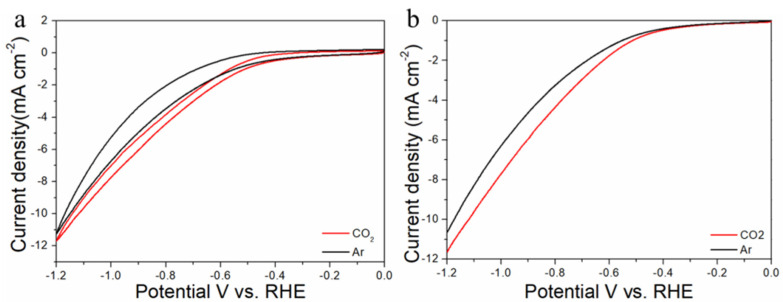
Cyclic voltammograms (**a**) and linear scan voltammograms (**b**) of Au_50_Cu_50_/C in Ar saturated and CO_2_-satruated 0.1 M KHCO_3_ solutions collected at a scan rate of 20 mV s^−1^.

**Figure 5 materials-15-05064-f005:**
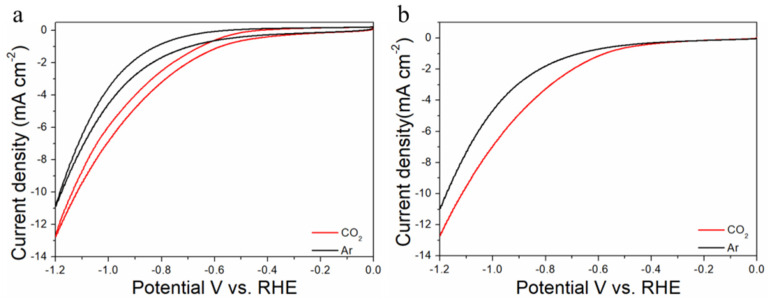
Cyclic voltammograms (**a**) and linear scan voltammograms (**b**) of Au_75_Cu_25_/C in Ar saturated and CO_2_-satruated 0.1 M KHCO_3_ solutions collected at a scan rate of 20 mV s^−1^.

**Figure 6 materials-15-05064-f006:**
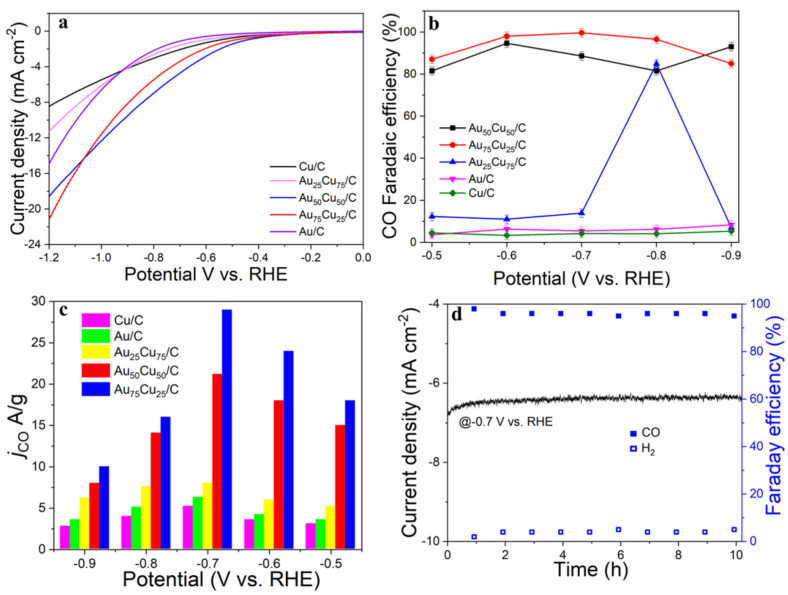
Performances of the Au_25_Cu_75_/C, Au_5__0_Cu_5__0_/C, Au_75_Cu_25_/C electrocatalysts for CO_2_ reduction with Cu/C and Au/C as controls. (**a**) The LSV in 0.1 M KHCO_3_ electrolyte saturated with CO_2_. (**b**) Faradaic efficiencies (%) at applied potential ranging from −0.5 V to −0.9V vs. RHE. (**c**) CO_2_ reduction current density (*j_CO_*) with all the carbonaceous products taken into account. (**d**) Stability test and FE(%) of CO and H_2_.

**Figure 7 materials-15-05064-f007:**
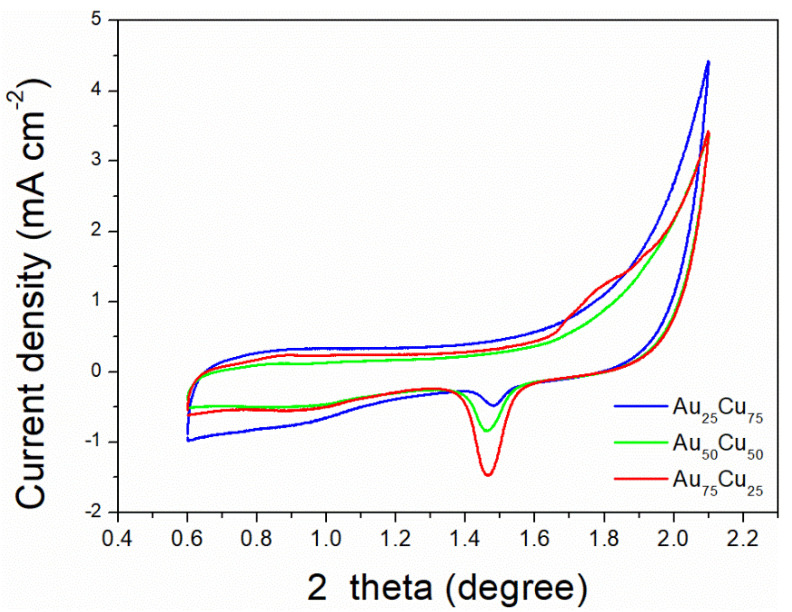
The ECSA of Au_25_Cu_75_/C, Au_50_Cu_50_/C and Au_75_Cu_25_/C. Cyclic voltammograms in 50 mM H_2_SO_4_, scan rate 50 mV s^−^^1^.

**Figure 8 materials-15-05064-f008:**
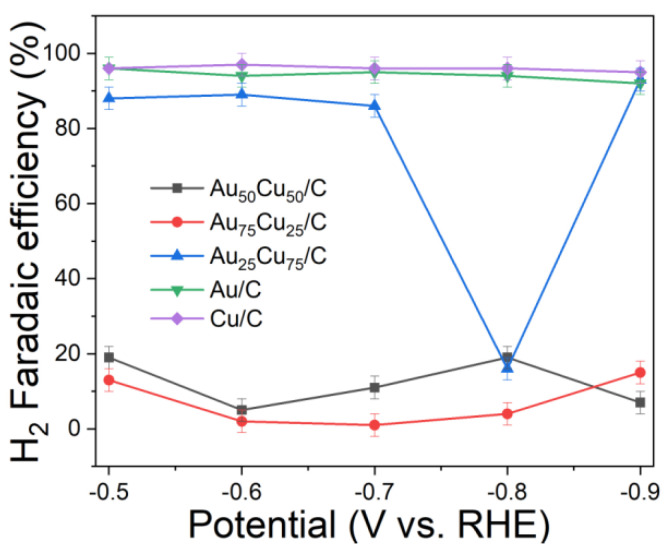
The H_2_ FE of Au_25_Cu_75_/C, Au_50_Cu_50_/C and Au_75_Cu_25_/C, Au/C and Cu/C.

**Figure 9 materials-15-05064-f009:**
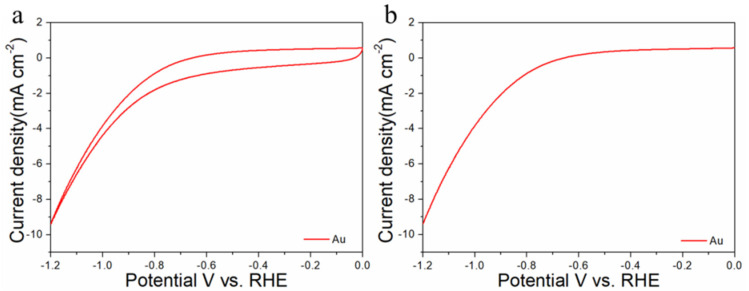
Cyclic voltammograms (**a**) and linear scan voltammograms (**b**) of Au/C in CO_2_-satruated 0.1 M KHCO_3_ solution collected at a scan rate of 20 mV s^−1^.

**Figure 10 materials-15-05064-f010:**
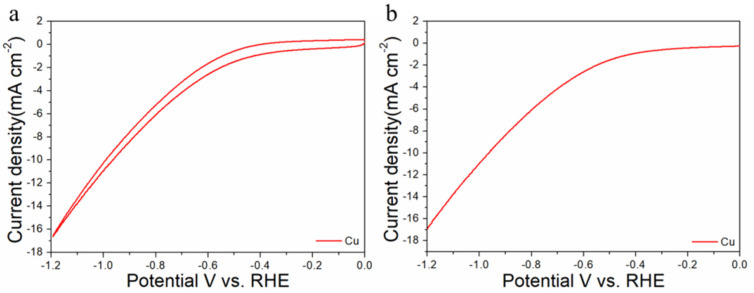
Cyclic voltammograms (**a**) and linear scan voltammograms (**b**) of Cu/C in CO_2_-satruated 0.1 M KHCO_3_ solution collected at a scan rate of 20 mV s^−1^.

**Figure 11 materials-15-05064-f011:**
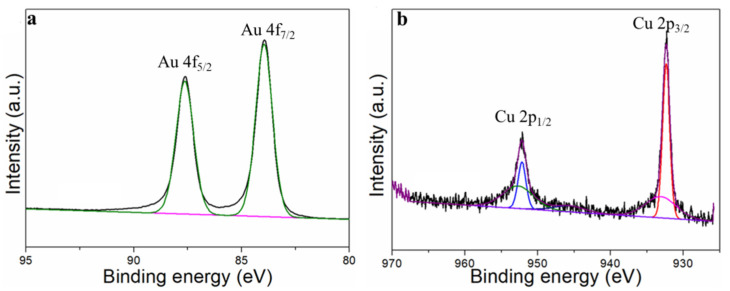
The XPS of Au_75_Cu_25_/C after the stability test. (**a**) Au and (**b**) for XPS data.

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
