# Peer review of "Strained Lattice Gold-Copper Alloy Nanoparticles for Efficient Carbon Dioxide Electroreduction"

_materials, 2022, doi:10.3390/ma15145064_

Round 1
Reviewer 1 Report
The article "Strained Lattice Gold-Copper Alloy Nanoparticles for Efficient Carbon Dioxide Electroreduction" by Fangfang Chang, Chenguang Wang, Xueli Wu, Yongpeng Liu, Juncai Wei, Zhengyu Bai, Lin Yang is an important original and interesting study on a topical modern problem - reducing the concentration of carbon dioxide in the atmosphere due to its recycling and can be accepted for publication in Materials after a few corrections.
- abbreviation RHE decoding appears only on line 130, the third occurrence in the text
- line 66. I would like to see the range of n index
- line 66-74 looks like a conclusion or annotation, but not like the final part of the introduction, which should contain a preview of the research
- line 88. "Several times" can mean quite a large range of values, clarification and explanation is required
- line 90, 98, 106. Check paragraph numbers
Legends in Fugures 2, 6 and 11 are completely unreadable and need to be improved. Tick labels should be also increased
Author Response
Point-by-point responses and revisions made in the manuscript (materials-1829719)
In this article, AunCu100-n/C nanoparticles were prepared and their electrocatalytic conversion of carbon dioxide (CO2) to CO has been investigated and the mechanism of conversion is explained. This study is well designed and executed and merit for a publication with some minor changes. Please address the following issues.
In the abstract please define the abbreviation RHE.
Reponses:
We have defined the abbreviation RHE in the abstract, see revised manuscript.
Please rephrase the following sentence.
“However, due to the number of electron transfer in CO2RR process, reaction pathway, hydrogen evolution reaction (HER) and unreasonable surface design of catalyst, etc. [16-18]” Incomplete sentence.
Reponses:
Thanks. We have revised it, see revised manuscript.
In the Materials and Methods Section, description for the synthesis of Cu NPs and AunCu100-n NPs are given. I did not see how Au75Cu25/C powder was fabricated. Please clarify this issue.
Reponses:
Line 97-98. Au75Cu25 catalyst was prepared under similar conditions when n(HAuCl4•xH2O):n(Cu(NO3)2) was 3:1.
In many places’ abbreviations are used but not defined in advances. Please fix deficiencies throughout the article.
Reponses:
Thanks. We have revised it, see revised manuscript.
In figure 2 (lower two frames) look blurry and not readable.
Reponses:
Thanks. We have revised it, see revised manuscript.
Reviewer 2 Report
In this article, AunCu100-n/C nanoparticles were prepared and their electrocatalytic conversion of carbon dioxide (CO2) to CO have been investigated and the mechanism of conversion is explained. This study is well designed and executed and merit for a publication with some minor changes. Please address the following issues.
In the abstract please define the abbreviation RHE.
Please rephrase the following sentence.
“However, due to the number of electron transfer in CO2RR process, reaction pathway, hydrogen evolution reaction (HER) and unreasonable surface design of catalyst, etc. [16-18]” Incomplete sentence.
In the Materials and Methods Section, description for the synthesis of Cu NPs and AunCu100-n NPs are given. I did not see how Au75Cu25/C powder was fabricated. Please clarify this issue.
In many places’ abbreviations are used but not defined in advances. Please fix deficiencies throughout the article.
In figure 2 (lower two frames) look blurry and not readable.
Author Response

(The authors gave the same response as above.)

Reviewer 3 Report
The manuscript entitled „Strained Lattice Gold-Copper Alloy Nanoparticles for Efficient Carbon 2 Dioxide Electroreduction” presents a current effort to reduce the utilization of the noble metal Au in a CO2 electrocatalyst through Cu alloying. The experimental results suggest that the Au/Cu ratio of 75:25 is necessary to obtain high-performing CO2 electroreduction.
The manuscript has been well-written and demonstrates a good result in the implementation of Au-Cu alloy in CO2 electroreduction. The material system is quite novel and might be of journal reader interest.
To be accepted for publication, a minor explanation should be given by the authors as follows:
1. A detailed discussion of the Au-Cu NPs synthesis and properties has not been adequately provided. The authors did not describe whether the Cu atoms incorporated into the Au crystal structure or formed an Au-Cu core-shell NPs structure. Line 143-144 describes “indicating that Cu successfully bound to the Au nanostructure, as shown in (Figure 1d-f)” but does not present clearly how Cu incorporates into Au. Moreover, the Au nanostructure is rather confusing unless there is a more detailed/clear explanation of what the authors mean by “nanostructure”.
2. Do the Au-Cu alloy NPs crystallize in face-centered tetragonal or face-centered cubic structures? And what is the XRD reference (ICDD or Crystallographic Open Database) number of the Au and Cu used by the authors?
3. A detailed method for the determination of the active surface area of Au-Cu NPs needs to be included in Materials and Methods section.
4. A discussion concerning what causes the high electrochemically active surface area is missing. Indeed, the higher the Au content (with Au 111 facet) in Au-Cu alloy NPs, the largest the electrochemically active surface area.
Author Response
Point-by-point responses and revisions made in the manuscript (materials-1829719)
The manuscript entitled “Strained Lattice Gold-Copper Alloy Nanoparticles for Efficient Carbon Dioxide Electroreduction” presents a current effort to reduce the utilization of the noble metal Au in a CO2 electrocatalyst through Cu alloying. The experimental results suggest that the Au/Cu ratio of 75:25 is necessary to obtain high-performing CO2 electroreduction.
The manuscript has been well-written and demonstrates a good result in the implementation of Au-Cu alloy in CO2 electroreduction. The material system is quite novel and might be of journal reader interest.
To be accepted for publication, a minor explanation should be given by the authors as follows:
- A detailed discussion of the Au-Cu NPs synthesis and properties has not been adequately provided. The authors did not describe whether the Cu atoms incorporated into the Au crystal structure or formed an Au-Cu core-shell NPs structure. Line 143-144 describes “indicating that Cu successfully bound to the Au nanostructure, as shown in (Figure 1d-f)” but does not present clearly how Cu incorporates into Au. Moreover, the Au nanostructure is rather confusing unless there is a more detailed/clear explanation of what the authors mean by “nanostructure”.
Reponses:
AunCu100-n NPs was prepared by galvanic replacement reaction assisted by Na2S2O3 in aqueous solution composed of HAuCl4 and copper NPs, where Na2S2O3 acted as an inhibitor of CuCl disproportionation caused by Cu/HAuCl4 substitution. The compositions of AunCu100-n NPs were controlled by adjusting the ratio of Cu NPs and HAuCl4. Line 143-144 description has been modified, see revised manuscript.
- Do the Au-Cu alloy NPs crystallize in face-centered tetragonal or face-centered cubic structures? And what is the XRD reference (ICDD or Crystallographic Open Database) number of the Au and Cu used by the authors?
Reponses:
Au-Cu alloy NPs crystallize in face-centered cubic structures.
- A detailed method for the determination of the active surface area of Au-Cu NPs needs to be included in Materials and Methods section.
Reponses:
The detailed method for the determination of the active surface area of Au-Cu NPs has been provided in Materials and Methods section.
- A discussion concerning what causes the high electrochemically active surface area is missing. Indeed, the higher the Au content (with Au 111 facet) in Au-Cu alloy NPs, the largest the electrochemically active surface area.
Reponses:
Thank you for your suggestion. Indeed, the value of the electrochemically active surface area is increased with Au content, which is related with facet and lattice strain. We have discussed it in the revised manuscript.